# Online Learning with Gaussian Payoffs and Side Observations

**Yifan Wu**[1]         **András György**[2]         **Csaba Szepesvári**[1]

[1]Dept. of Computing Science          [2]Dept. of Electrical and Electronic Engineering
University of Alberta                          Imperial College London
`{ywu12,szepesva}@ualberta.ca`          `a.gyorgy@imperial.ac.uk`

## Abstract

We consider a sequential learning problem with Gaussian payoffs and side observations: after selecting an action $i$, the learner receives information about the payoff of every action $j$ in the form of Gaussian observations whose mean is the same as the mean payoff, but the variance depends on the pair $(i, j)$ (and may be infinite). The setup allows a more refined information transfer from one action to another than previous partial monitoring setups, including the recently introduced graph-structured feedback case. For the first time in the literature, we provide non-asymptotic problem-dependent lower bounds on the regret of any algorithm, which recover existing asymptotic problem-dependent lower bounds and finite-time minimax lower bounds available in the literature. We also provide algorithms that achieve the problem-dependent lower bound (up to some universal constant factor) or the minimax lower bounds (up to logarithmic factors).

## 1   Introduction

Online learning in stochastic environments is a sequential decision problem where in each time step a learner chooses an action from a given finite set, observes some random feedback and receives a random payoff. Several feedback models have been considered in the literature: The simplest is the full information case where the learner observes the payoff of all possible actions at the end of every round. A popular setup is the case of bandit feedback, where the learner only observes its own payoff and receives no information about the payoff of other actions [1]. Recently, several papers considered a more refined setup, called graph-structured feedback, that interpolates between the full-information and the bandit case: here the feedback structure is described by a (possibly directed) graph, and choosing an action reveals the payoff of all actions that are connected to the selected one, including the chosen action itself. This problem, motivated for example by social networks, has been studied extensively in both the adversarial [2, 3, 4, 5] and the stochastic cases [6, 7]. However, most algorithms presented heavily depend on the self-observability assumption, that is, that the payoff of the selected action can be observed. Removing this self-loop assumption leads to the so-called partial monitoring case [5]. In the absolutely general partial monitoring setup the learner receives some general feedback that depends on its choice (and the environment), with some arbitrary (but known) dependence [8, 9]. While the partial monitoring setup covers all other problems, its analysis has concentrated on the finite case where both the set of actions and the set of feedback signals are finite [8, 9], which is in contrast to the standard full information and bandit settings where the feedback is typically assumed to be real-valued. To our knowledge there are only a few exceptions to this case: in [5], graph-structured feedback is considered without the self-loop assumption, while continuous action spaces are considered in [10] and [11] with special feedback structure (linear and censored observations, resp.).

In this paper we consider a generalization of the graph-structured feedback model that can also be viewed as a general partial monitoring model with real-valued feedback. We assume that selecting

an action $i$ the learner can observe a random variable $X_{ij}$ for each action $j$ whose mean is the same as the payoff of $j$, but its variance $\sigma_{ij}^2$ depends on the pair $(i, j)$. For simplicity, throughout the paper we assume that all the payoffs and the $X_{ij}$ are Gaussian. While in the graph-structured feedback case one either has observation on an action or not, but the observation always gives the same amount of information, our model is more refined: Depending on the value of $\sigma_{ij}^2$, the information can be of different quality. For example, if $\sigma_{ij}^2 = \infty$, trying action $i$ gives no information about action $j$. In general, for any $\sigma_{ij}^2 < \infty$, the value of the information depends on the time horizon $T$ of the problem: when $\sigma_{ij}^2$ is large relative to $T$ (and the payoff differences of the actions) essentially no information is received, while a small variance results in useful observations.

After defining the problem formally in Section 2, we provide non-asymptotic problem-dependent lower bounds in Section 3, which depend on the distribution of the observations through their mean payoffs and variances. To our knowledge, these are the first such bounds presented for any stochastic partial monitoring problem beyond the full-information setting: previous work either presented asymptotic problem-dependent lower bounds (e.g., [12, 7]), or finite-time minimax bounds (e.g., [9, 3, 5]). Our bounds can recover all previous bounds up to some universal constant factors not depending on the problem. In Section 4, we present two algorithms with finite-time performance guarantees for the case of graph-structured feedback without the self-observability assumption. While due to their complicated forms it is hard to compare our finite-time upper and lower bounds, we show that our first algorithm achieves the asymptotic problem-dependent lower bound up to problem-independent multiplicative factors. Regarding the minimax regret, the hardness ($\widetilde{\Theta}(T^{1/2})$ or $\widetilde{\Theta}(T^{2/3})$ regret[1]) of partial monitoring problems is characterized by their global/local observability property [9] or, in case of the graph-structured feedback model, by their strong/weak observability property [5]. In the same section we present another algorithm that achieves the minimax regret (up to logarithmic factors) under both strong and weak observability, and achieves an $O(\log^{3/2} T)$ problem-dependent regret. Earlier results for the stochastic graph-structured feedback problems [6, 7] provided only asymptotic problem-dependent lower bounds and performance bounds that did not match the asymptotic lower bounds or the minimax rate up to constant factors. A related combinatorial partial monitoring problem with linear feedback was considered in [10], where the presented algorithm was shown to satisfy both an $\widetilde{O}(T^{2/3})$ minimax bound and a logarithmic problem dependent bound. However, the dependence on the problem structure in that paper is not optimal, and, in particular, the paper does not achieve the $O(\sqrt{T})$ minimax bound for easy problems. Finally, we draw conclusions and consider some interesting future directions in Section 5. Proofs can be found in the long version of this paper [13].

## 2 Problem Formulation

Formally, we consider an online learning problem with *Gaussian payoffs and side observations*: Suppose a learner has to choose from $K$ actions in every round. When choosing an action, the learner receives a random payoff and also some side observations corresponding to other actions. More precisely, each action $i \in [K] = \{1, \ldots, K\}$ is associated with some parameter $\theta_i$, and the payoff $Y_{t,i}$ to action $i$ in round $t$ is normally distributed random variable with mean $\theta_i$ and variance $\sigma_{ii}^2$, while the learner observes a $K$-dimensional Gaussian random vector $X_{t,i}$ whose $j$th coordinate is a normal random variable with mean $\theta_j$ and variance $\sigma_{ij}^2$ (we assume $0 \le \sigma_{ij} \le \infty$) and the coordinates of $X_{t,i}$ are independent of each other. We assume the following: (i) the random variables $(X_t, Y_t)_t$ are independent for all $t$; (ii) the parameter vector $\theta$ is unknown to the learner but the variance matrix $\Sigma = (\sigma_{ij}^2)_{i,j \in [K]}$ is known in advance; (iii) $\theta \in [0, D]^K$ for some $D > 0$; (iv) $\min_{i \in [K]} \sigma_{ij} \le \sigma < \infty$ for all $j \in [K]$, that is, the expected payoff of each action can be observed.

The goal of the learner is to maximize its payoff or, in other words, minimize the expected regret

$$R_T = T \max_{i \in [K]} \theta_i - \sum_{t=1}^{T} \mathbb{E}\left[Y_{t,i_t}\right]$$

where $i_t$ is the action selected by the learner in round $t$. Note that the problem encompasses several common feedback models considered in online learning (modulo the Gaussian assumption), and makes it possible to examine more delicate observation structures:

**Full information:** $\sigma_{ij} = \sigma_j < \infty$ for all $i, j \in [K]$.

**Bandit:** $\sigma_{ii} < \infty$ and $\sigma_{ij} = \infty$ for all $i \neq j \in [K]$.

**Partial monitoring with feedback graphs [5]:** Each action $i \in [K]$ is associated with an observation set $S_i \subset [K]$ such that $\sigma_{ij} = \sigma_j < \infty$ if $j \in S_i$ and $\sigma_{ij} = \infty$ otherwise.

We will call the *uniform variance* version of these problems when all the finite $\sigma_{ij}$ are equal to some $\sigma \geq 0$. Some interesting features of the problem can be seen when considering the *generalized full information* case , when all entries of $\Sigma$ are finite. In this case, the greedy algorithm, which estimates the payoff of each action by the average of the corresponding observed samples and selects the one with the highest average, achieves at most a constant regret for any time horizon $T$.[2] On the other hand, the constant can be quite large: in particular, when the variance of some observations are large relative to the gaps $d_j = \max_i \theta_i - \theta_j$, the situation is rather similar to a partial monitoring setup for a smaller, finite time horizon. In this paper we are going to analyze this problem and present algorithms and lower bounds that are able to "interpolate" between these cases and capture the characteristics of the different regimes.

## 2.1 Notation

Define $C_T^{\mathbb{N}} = \{c \in \mathbb{N}^K : c_i \geq 0, \sum_{i \in [K]} c_i = T\}$ and let $N(T) \in C_T^{\mathbb{N}}$ denote the number of plays over all actions taken by some algorithm in $T$ rounds. Also let $C_T^{\mathbb{R}} = \{c \in \mathbb{R}^K : c_i \geq 0, \sum_{i \in [K]} c_i = T\}$. We will consider environments with different expected payoff vectors $\theta \in \Theta$, but the variance matrix $\Sigma$ will be fixed. Therefore, an environment can be specified by $\theta$; oftentimes, we will explicitly denote the dependence of different quantities on $\theta$: The probability and expectation functionals under environment $\theta$ will be denoted by $\Pr(\cdot; \theta)$ and $\mathbb{E}[\cdot; \theta]$, respectively. Furthermore, let $i_j(\theta)$ be the $j$th best action (ties are broken arbitrarily, i.e., $\theta_{i_1} \geq \theta_{i_2} \geq \cdots \geq \theta_{i_K}$) and define $d_i(\theta) = \theta_{i_1(\theta)} - \theta_i$ for any $i \in [K]$. Then the expected regret under environment $\theta$ is $R_T(\theta) = \sum_{i \in [K]} \mathbb{E}[N_i(T); \theta] d_i(\theta)$. For any action $i \in [K]$, let $S_i = \{j \in [K] : \sigma_{ij} < \infty\}$ denote the set of actions whose parameter $\theta_j$ is observable by choosing action $i$. Throughout the paper, $\log$ denotes the natural logarithm and $\dot{\Delta}^n$ denotes the $n$-dimensional simplex for any positive integer $n$.

## 3 Lower Bounds

The aim of this section is to derive generic, problem-dependent lower bounds to the regret, which are also able to provide minimax lower bounds. The hardness in deriving such bounds is that for any fixed $\theta$ and $\Sigma$, the dumb algorithm that always selects $i_1(\theta)$ achieves zero regret (obviously, the regret of this algorithm is linear for any $\theta'$ with $i_1(\theta) \neq i_1(\theta')$), so in general it is not possible to give a lower bound for a single instance. When deriving asymptotic lower bounds, this is circumvented by only considering *consistent* algorithms whose regret is sub-polynomial for any problem [12]. However, this asymptotic notion of consistency is not applicable to finite-horizon problems. Therefore, following ideas of [14], for any problem we create a family of *related* problems (by perturbing the mean payoffs) such that if the regret of an algorithm is "too small" in one of the problems than it will be "large" in another one, while it still depends on the original problem parameters (note that deriving minimax bounds usually only involves perturbing certain special "worst-case" problems).

As a warm-up, and to show the reader what form of a lower bound can be expected, first we present an asymptotic lower bound for the uniform-variance version of the problem of *partial monitoring with feedback graphs*. The result presented below is an easy consequence of [12], hence its proof is omitted. An algorithm is said to be *consistent* if $\sup_{\theta \in \Theta} R_T(\theta) = o(T^\gamma)$ for every $\gamma > 0$. Now assume for simplicity that there is a unique optimal action in environment $\theta$, that is, $\theta_{i_1(\theta)} > \theta_i$ for all $i \neq i_1$ and let

$$C_\theta = \left\{ c \in [0, \infty)^K : \sum_{i:j \in S_i} c_i \geq \frac{2\sigma^2}{d_j^2(\theta)} \text{ for all } j \neq i_1(\theta), \sum_{i:i_1(\theta) \in S_i} c_i \geq \frac{2\sigma^2}{d_{i_2(\theta)}^2(\theta)} \right\}.$$

Then, for any consistent algorithm and for any $\theta$ with $\theta_{i_1(\theta)} > \theta_{i_2(\theta)}$,

$$\liminf_{T \to \infty} \frac{R_T(\theta)}{\log T} \geq \inf_{c \in C_\theta} \langle c, d(\theta) \rangle . \tag{1}$$

Note that the right hand side of (1) is 0 for any *generalized full information* problem (recall that the expected regret is bounded by a constant for such problems), but it is a finite positive number for other problems. Similar bounds have been provided in [6, 7] for graph-structured feedback with self-observability (under non-Gaussian assumptions on the payoffs). In the following we derive finite time lower bounds that are also able to replicate this result.

## 3.1 A General Finite Time Lower Bound

First we derive a general lower bound. For any $\theta, \theta' \in \Theta$ and $q \in \Delta^{|C_T^{\mathbb{N}}|}$, define $f(\theta, q, \theta')$ as

$$f(\theta, q, \theta') = \inf_{q' \in \Delta^{|C_T^{\mathbb{N}}|}} \sum_{a \in C_T^{\mathbb{N}}} q'(a) \langle a, d(\theta') \rangle$$

$$\text{such that} \sum_{a \in C_T^{\mathbb{N}}} q(a) \log \frac{q(a)}{q'(a)} \leq \sum_{i \in [K]} \left( I_i(\theta, \theta') \sum_{a \in C_T^{\mathbb{N}}} q(a) a_i \right) ,$$

where $I_i(\theta, \theta')$ is the KL-divergence between $X_{t,i}(\theta)$ and $X_{t,i}(\theta')$, given by $I_i(\theta, \theta') = \text{KL}(X_{t,i}(\theta); X_{t,i}(\theta')) = \sum_{j=1}^{K}(\theta_j - \theta_j')^2/2\sigma_{ij}^2$. Clearly, $f(\theta, q, \theta')$ is a lower bound on $R_T(\theta')$ for any algorithm for which the distribution of $N(T)$ is $q$. The intuition behind the allowed values of $q'$ is that we want $q'$ to be as similar to $q$ as the environments $\theta$ and $\theta'$ look like for the algorithm (through the feedback $(X_{t,i_t})_t$). Now define

$$g(\theta, c) = \inf_{q \in \Delta^{|C_T^{\mathbb{N}}|}} \sup_{\theta' \in \Theta} f(\theta, q, \theta'), \qquad \text{such that} \sum_{a \in C_T^{\mathbb{N}}} q(a) a = c \in C_T^{\mathbb{R}}.$$

$g(\theta, c)$ is a lower bound of the worst-case regret of any algorithm with $\mathbb{E}[N(T); \theta] = c$. Finally, for any $x > 0$, define

$$b(\theta, x) = \inf_{c \in C_{\theta, x}} \langle c, d(\theta) \rangle \qquad \text{where} \quad C_{\theta, x} = \{ c \in C_T^{\mathbb{R}} ; g(\theta, c) \leq x \}.$$

Here $C_{\theta, B}$ contains all the possible values of $\mathbb{E}[N(T); \theta]$ that can be achieved by some algorithm whose lower bound $g$ on the worst-case regret is smaller than $x$. These definitions give rise to the following theorem:

**Theorem 1.** *Given any $B > 0$, for any algorithm such that $\sup_{\theta' \in \Theta} R_T(\theta') \leq B$, we have, for any environment $\theta \in \Theta$, $R_T(\theta) \geq b(\theta, B)$.*

**Remark 2.** If $B$ is picked as the minimax value of the problem given the observation structure $\Sigma$, the theorem states that for any minimax optimal algorithm the expected regret for a certain $\theta$ is lower bounded by $b(\theta, B)$.

## 3.2 A Relaxed Lower Bound

Now we introduce a relaxed but more interpretable version of the finite-time lower bound of Theorem 1, which can be shown to match the asymptotic lower bound (1). The idea of deriving the lower bound is the following: instead of ensuring that the algorithm performs well in the most adversarial environment $\theta'$, we consider a set of "bad" environments and make sure that the algorithm performs well on them, where each "bad" environment $\theta'$ is the most adversarial one by only perturbing one coordinate $\theta_i$ of $\theta$.

However, in order to get meaningful finite-time lower bounds, we need to perturb $\theta$ more carefully than in the case of asymptotic lower bounds. The reason for this is that for any sub-optimal action $i$, if $\theta_i$ is very close to $\theta_{i_1(\theta)}$, then $\mathbb{E}[N_i(T); \theta]$ is not necessarily small for a good algorithm for $\theta$. If it is small, one can increase $\theta_i$ to obtain an environment $\theta'$ where $i$ is the best action and the algorithm performs bad; otherwise, when $\mathbb{E}[N_i(T); \theta]$ is large, we need to decrease $\theta_i$ to make the

algorithm perform badly in $\theta'$. Moreover, when perturbing $\theta_i$ to be better than $\theta_{i_1(\theta)}$, we cannot make $\theta'_i - \theta_{i_1(\theta)}$ arbitrarily small as in asymptotic lower-bound arguments, because when $\theta'_i - \theta_{i_1(\theta)}$ is small, large $\mathbb{E}\left[N_{i_1(\theta)}; \theta'\right]$, and not necessarily large $\mathbb{E}\left[N_i(T); \theta'\right]$, may also lead to low finite-time regret in $\theta'$. In the following we make this argument precise to obtain an interpretable lower bound.

### 3.2.1 Formulation

We start with defining a subset of $C_T^{\mathbb{R}}$ that contains the set of "reasonable" values for $\mathbb{E}[N(T); \theta]$. For any $\theta \in \Theta$ and $B > 0$, let

$$C'_{\theta,B} = \left\{ c \in C_T^{\mathbb{R}} \ : \ \sum_{j=1}^{K} \frac{c_j}{\sigma_{ji}^2} \geq m_i(\theta, B) \ \text{for all } i \in [K] \right\}$$

where $m_i$, the minimum sample size required to distinguish between $\theta_i$ and its worst-case perturbation, is defined as follows: For $i \neq i_1$, if $\theta_{i_1} = D,$[3] then $m_i(\theta, B) = 0$. Otherwise let

$$m_{i,+}(\theta, B) = \max_{\epsilon \in (d_i(\theta), D - \theta_i]} \frac{1}{\epsilon^2} \log \frac{T(\epsilon - d_i(\theta))}{8B},$$

$$m_{i,-}(\theta, B) = \max_{\epsilon \in (0, \theta_i]} \frac{1}{\epsilon^2} \log \frac{T(\epsilon + d_i(\theta))}{8B},$$

and let $\epsilon_{i,+}$ and $\epsilon_{i,-}$ denote the value of $\epsilon$ achieving the maximum in $m_{i,+}$ and $m_{i,-}$, respectively. Then, define

$$m_i(\theta, B) = \begin{cases} m_{i,+}(\theta, B) & \text{if } d_i(\theta) \geq 4B/T; \\ \min\{m_{i,+}(\theta, B), m_{i,-}(\theta, B)\} & \text{if } d_i(\theta) < 4B/T. \end{cases}$$

For $i = i_1$, then $m_{i_1}(\theta, B) = 0$ if $\theta_{i_2(\theta)} = 0$, else the definitions for $i \neq i_1$ change by replacing $d_i(\theta)$ with $d_{i_2(\theta)}(\theta)$ (and switching the + and − indices):

$$m_{i_1(\theta),-}(\theta, B) = \max_{\epsilon \in (d_{i_2(\theta)}(\theta), \theta_{i_1(\theta)}]} \frac{1}{\epsilon^2} \log \frac{T(\epsilon - d_{i_2(\theta)}(\theta))}{8B},$$

$$m_{i_1(\theta),+}(\theta, B) = \max_{\epsilon \in (0, D - \theta_{i_1(\theta)}]} \frac{1}{\epsilon^2} \log \frac{T(\epsilon + d_{i_2(\theta)}(\theta))}{8B}$$

where $\epsilon_{i_1(\theta),-}$ and $\epsilon_{i_1(\theta),+}$ are the maximizers for $\epsilon$ in the above expressions. Then, define

$$m_{i_1(\theta)}(\theta, B) = \begin{cases} m_{i_1(\theta),-}(\theta, B) & \text{if } d_{i_2(\theta)}(\theta) \geq 4B/T; \\ \min\{m_{i_1(\theta),+}(\theta, B), m_{i_1(\theta),-}(\theta, B)\} & \text{if } d_{i_2(\theta)}(\theta) < 4B/T. \end{cases}$$

Note that $\epsilon_{i,+}$ and $\epsilon_{i,-}$ can be expressed in closed form using the Lambert $W : \mathbb{R} \to \mathbb{R}$ function satisfying $W(x)e^{W(x)} = x$: for any $i \neq i_1(\theta)$,

$$\begin{aligned}
\epsilon_{i,+} &= \min\left\{ D - \theta_i, \, 8\sqrt{e}Be^{W\left(\frac{d_i(\theta)T}{16\sqrt{e}B}\right)}/T + d_i(\theta) \right\}, \\
\epsilon_{i,-} &= \min\left\{ \theta_i, \, 8\sqrt{e}Be^{W\left(-\frac{d_i(\theta)T}{16\sqrt{e}B}\right)}/T - d_i(\theta) \right\},
\end{aligned} \tag{2}$$

and similar results hold for $i = i_1$, as well.

Now we can give the main result of this section, a simplified version of Theorem 1:

**Corollary 3.** *Given $B > 0$, for any algorithm such that $\sup_{\lambda \in \Theta} R_T(\lambda) \leq B$, we have, for any environment $\theta \in \Theta$, $R_T(\theta) \geq b'(\theta, B) = \min_{c \in C'_{\theta,B}} \langle c, d(\theta) \rangle$.*

Next we compare this bound to existing lower bounds.

### 3.2.2 Comparison to the Asymptotic Lower Bound (1)

Now we will show that our finite time lower bound in Corollary 3 matches the asymptotic lower bound in (1) up to some constants. Pick $B = \alpha T^\beta$ for some $\alpha > 0$ and $0 < \beta < 1$. For simplicity, we only consider $\theta$ which is "away from" the boundary of $\Theta$ (so that the minima in (2) are

achieved by the second terms) and has a unique optimal action. Then, for $i \neq i_1(\theta)$, it is easy to show that $\epsilon_{i,+} = \frac{d_i(\theta)}{2W(d_i(\theta)T^{1-\beta}/(16\alpha\sqrt{e}))} + d_i(\theta)$ by (2) and $m_i(\theta, B) = \frac{1}{\epsilon_{i,+}^2} \log \frac{T(\epsilon_{i,+} - d_i(\theta))}{8B}$ for large enough $T$. Then, using the fact that $\log x - \log \log x \leq W(x) \leq \log x$ for $x \geq e$, it follows that $\lim_{T \to \infty} m_i(\theta, B)/\log T = (1 - \beta)/d_i^2(\theta)$, and similarly we can show that $\lim_{T \to \infty} m_{i_1(\theta)}(\theta, B)/\log T = (1 - \beta)/d_{i_2(\theta)}^2(\theta)$. Thus, $C'_{\theta,B} \to \frac{(1-\beta)\log T}{2} C_\theta$, under the assumptions of (1), as $T \to \infty$. This implies that Corollary 3 matches the asymptotic lower bound of (1) up to a factor of $(1 - \beta)/2$.

### 3.2.3 Comparison to Minimax Bounds

Now we will show that our $\theta$-dependent finite-time lower bound reproduces the minimax regret bounds of [2] and [5], except for the generalized full information case.

The minimax bounds depend on the following notion of observability: An action $i$ is *strongly observable* if either $i \in S_i$ or $[K] \setminus \{i\} \subset \{j : i \in S_j\}$. $i$ is *weakly observable* if it is not strongly observable but there exists $j$ such that $i \in S_j$ (note that we already assumed the latter condition for all $i$). Let $\mathcal{W}(\Sigma)$ be the set of all weakly observable actions. $\Sigma$ is said to be strongly observable if $\mathcal{W}(\Sigma) = \emptyset$. $\Sigma$ is weakly observable if $\mathcal{W}(\Sigma) \neq \emptyset$.

Next we will define two key qualities introduced by [2] and [5] that characterize the hardness of a problem instance with feedback structure $\Sigma$: A set $A \subset [K]$ is called an independent set if for any $i \in A$, $S_i \cap A \subset \{i\}$. The *independence number* $\kappa(\Sigma)$ is defined as the cardinality of the largest independent set. For any pair of subsets $A, A' \subset [K]$, $A$ is said to be *dominating* $A'$ if for any $j \in A'$ there exists $i \in A$ such that $j \in S_i$. The *weak domination number* $\rho(\Sigma)$ is defined as the cardinality of the smallest set that dominates $\mathcal{W}(\Sigma)$.

**Corollary 4.** *Assume that $\sigma_{ij} = \infty$ for some $i, j \in [K]$, that is, we are not in the generalized full information case. Then,*

*(i) if $\Sigma$ is strongly observable, with $B = \alpha\sigma\sqrt{\kappa(\Sigma)T}$ for some $\alpha > 0$, we have*
$$\sup_{\theta \in \Theta} b'(\theta, B) \geq \frac{\sigma\sqrt{\kappa(\Sigma)T}}{64e\alpha} \text{ for } T \geq 64e^2\alpha^2\sigma^2\kappa(\Sigma)^3/D^2.$$

*(ii) If $\Sigma$ is weakly observable, with $B = \alpha(\rho(\Sigma)D)^{1/3}(\sigma T)^{2/3}\log^{-2/3} K$ for some $\alpha > 0$, we have $\sup_{\theta \in \Theta} b'(\theta, B) \geq \frac{(\rho(\Sigma)D)^{1/3}(\sigma T)^{2/3}\log^{-2/3} K}{51200e^2\alpha^2}$.*

**Remark 5.** In Corollary 4, picking $\alpha = \frac{1}{8\sqrt{e}}$ for strongly observable $\Sigma$ and $\alpha = \frac{1}{73}$ for weakly observable $\Sigma$ gives formal minimax lower bounds: (i) If $\Sigma$ is strongly observable, for any algorithm we have $\sup_{\theta \in \Theta} R_T(\theta) \geq \frac{\sigma\sqrt{\kappa(\Sigma)T}}{8\sqrt{e}}$ for $T \geq e\sigma^2\kappa(\Sigma)^3/D^2$. (ii) If $\Sigma$ is weakly observable, for any algorithm we have $\sup_{\theta \in \Theta} R_T(\theta) \geq \frac{(\rho(\Sigma)D)^{1/3}(\sigma T)^{2/3}}{73\log^{2/3} K}$.

## 4 Algorithms

In this section we present two algorithms and their finite-time analysis for the uniform variance version of our problem (where $\sigma_{ij}$ is either $\sigma$ or $\infty$). The upper bound for the first algorithm matches the asymptotic lower bound in (1) up to constants. The second algorithm achieves the minimax lower bounds of Corollary 4 up to logarithmic factors, as well as $O(\log^{3/2} T)$ problem-dependent regret. In the problem-dependent upper bounds of both algorithms, we assume that the optimal action is unique, that is, $d_{i_2(\theta)}(\theta) > 0$.

### 4.1 An Asymptotically Optimal Algorithm

Let $c(\theta) = \operatorname{argmin}_{c \in C_\theta} \langle c, d(\theta) \rangle$; note that increasing $c_{i_1(\theta)}(\theta)$ does not change the value of $\langle c, d(\theta) \rangle$ (since $d_{i_1(\theta)}(\theta) = 0$), so we take the minimum value of $c_{i_1(\theta)}(\theta)$ in this definition. Let $n_i(t) = \sum_{s=1}^{t-1} \mathbb{I}\{i \in S_{i_s}\}$ be the number of observations for action $i$ before round $t$ and $\hat{\theta}_{t,i}$ be the empirical estimate of $\theta_i$ based on the first $n_i(t)$ observations. Let $N_i(t) = \sum_{s=1}^{t-1} \mathbb{I}\{i_s = i\}$ be the number of plays for action $i$ before round $t$. Note that this definition of $N_i(t)$ is different from that in the previous sections since it excludes round $t$.

**Algorithm 1**

1: Inputs: $\Sigma$, $\alpha$, $\beta : \mathbb{N} \to [0, \infty)$.
2: For $t = 1, ..., K$, observe each action $i$ at least once by playing $i_t$ such that $t \in S_{i_t}$.
3: Set exploration count $n_e(K + 1) = 0$.
4: **for** $t = K + 1, K + 2, ...$ **do**
5:    **if** $\frac{N(t)}{4\alpha \log t} \in C_{\hat{\theta}_t}$ **then**
6:       Play $i_t = i_1(\hat{\theta}_t)$.
7:       Set $n_e(t + 1) = n_e(t)$.
8:    **else**
9:       **if** $\min_{i \in [K]} n_i(t) < \beta(n_e(t))/K$ **then**
10:          Play $i_t$ such that $\operatorname{argmin}_{i \in [K]} n_i(t) \in S_{i_t}$.
11:       **else**
12:          Play $i_t$ such that $N_i(t) < c_i(\hat{\theta}_t)4\alpha \log t$.
13:       **end if**
14:       Set $n_e(t + 1) = n_e(t) + 1$.
15:    **end if**
16: **end for**

Our first algorithm is presented in Algorithm 1. The main idea, coming from [15], is that by forcing exploration over all actions, the solution $c(\theta)$ of the linear program can be well approximated while paying a constant price. This solves the main difficulty that, without getting enough observations on each action, we may not have good enough estimates for $d(\theta)$ and $c(\theta)$. One advantage of our algorithm compared to that of [15] is that we use a nondecreasing, sublinear exploration schedule $\beta(n)$ ($\beta : \mathbb{N} \to [0, \infty)$) instead of a constant rate $\beta(n) = \beta n$. This resolves the problem that, to achieve asymptotically optimal performance, some parameter of the algorithm needs to be chosen according to $d_{\min}(\theta)$ as in [15]. The expected regret of Algorithm 1 is upper bounded as follows:

**Theorem 6.** *For any $\theta \in \Theta$, $\epsilon > 0$, $\alpha > 2$ and any non-decreasing $\beta(n)$ that satisfies $0 \le \beta(n) \le n/2$ and $\beta(m + n) \le \beta(m) + \beta(n)$ for $m, n \in \mathbb{N}$,*

$$R_T(\theta) \le \big(2K + 2 + 4K/(\alpha - 2)\big)d_{\max}(\theta) + 4Kd_{\max}(\theta) \sum_{s=0}^{T} \exp\Big( -\frac{\beta(s)\epsilon^2}{2K\sigma^2} \Big)$$

$$+ 2d_{\max}(\theta)\beta\Big(4\alpha \log T \sum_{i \in [K]} c_i(\theta, \epsilon) + K\Big) + 4\alpha \log T \sum_{i \in [K]} c_i(\theta, \epsilon)d_i(\theta).$$

*where $c_i(\theta, \epsilon) = \sup\{c_i(\theta') : |\theta'_j - \theta_j| \le \epsilon \text{ for all } j \in [K]\}$.*

Further specifying $\beta(n)$ and using the continuity of $c(\theta)$ around $\theta$, it immediately follows that Algorithm 1 achieves asymptotically optimal performance:

**Corollary 7.** *Suppose the conditions of Theorem 6 hold. Assume, furthermore, that $\beta(n)$ satisfies $\beta(n) = o(n)$ and $\sum_{s=0}^{\infty} \exp\Big(-\frac{\beta(s)\epsilon^2}{2K\sigma^2}\Big) < \infty$ for any $\epsilon > 0$, then for any $\theta$ such that $c(\theta)$ is unique,*

$$\limsup_{T \to \infty} R_T(\theta)/\log T \le 4\alpha \inf_{c \in C(\theta)} \langle c, d(\theta) \rangle .$$

Note that any $\beta(n) = an^b$ with $a \in (0, \frac{1}{2}]$, $b \in (0, 1)$ satisfies the requirements in Theorem 6 and Corollary 7. Also note that the algorithms presented in [6, 7] do not achieve this asymptotic bound.

## 4.2 A Minimax Optimal Algorithm

Next we present an algorithm achieving the minimax bounds. For any $A, A' \subset [K]$, let $c(A, A') = \operatorname{argmax}_{c \in \Delta^{|A|}} \min_{i \in A'} \sum_{j:i \in S_j} c_j$ (ties are broken arbitrarily) and $m(A, A') = \min_{i \in A'} \sum_{j:i \in S_j} c_j(A, A')$. For any $A \subset [K]$ and $|A| \ge 2$, let $A^{\mathcal{S}} = \{i \in A : \exists j \in A, i \in S_j\}$ and $A^{\mathcal{W}} = A - A^{\mathcal{S}}$. Furthermore, let $g_{r,i}(\delta) = \sigma \sqrt{\frac{2\log(8K^2 r^3/\delta)}{n_i(r)}}$ where $n_i(r) = \sum_{s=1}^{r-1} i_{s,i}$ and $\hat{\theta}_{r,i}$ be the empirical estimate of $\theta_i$ based on first $n_i(r)$ observations (i.e., the average of the samples).

The algorithm is presented in Algorithm 2. It follows a successive elimination process: it explores all possibly optimal actions (called "good actions" later) based on some confidence intervals until only one action remains. While doing exploration, the algorithm first tries to explore the good actions by only using good ones. However, due to weak observability, some good actions might have to be explored by actions that have already been eliminated. To control this exploration-exploitation trade off, we use a sublinear function $\gamma$ to control the exploration of weakly observable actions.

In the following we present high-probability bounds on the performance of the algorithm, so, with a slight abuse of notation, $R_T(\theta)$ will denote the regret without expectation in the rest of this section.

**Algorithm 2**

1: Inputs: $\Sigma$, $\delta$.
2: Set $t_1 = 0$, $A_1 = [K]$.
3: **for** $r = 1, 2, \ldots$ **do**
4:     Let $\alpha_r = \min_{1 \le s \le r, A_s^{\mathcal{W}} \ne \emptyset} m([K], A_s^{\mathcal{W}})$ and $\gamma(r) = (\sigma \alpha_r t_r / D)^{2/3}$. (Define $\alpha_r = 1$ if $A_s^{\mathcal{W}} = \emptyset$ for all $1 \le s \le r$.)
5:     **if** $A_r^{\mathcal{W}} \ne \emptyset$ and $\min_{i \in A_r^{\mathcal{W}}} n_i(r) < \min_{i \in A_r^{\mathcal{S}}} n_i(r)$ and $\min_{i \in A_r^{\mathcal{W}}} n_i(r) < \gamma(r)$ **then**
6:         Set $c_r = c([K], A_r^{\mathcal{W}})$.
7:     **else**
8:         Set $c_r = c(A_r, A_r^{\mathcal{S}})$.
9:     **end if**
10:     Play $i_r = \lceil c_r \cdot \|c_r\|_0 \rceil$ and set $t_{r+1} \leftarrow t_r + \|i_r\|_1$.
11:     $A_{r+1} \leftarrow \{i \in A_r : \hat{\theta}_{r+1,i} + g_{r+1,i}(\delta) \ge \max_{j \in A_r} \hat{\theta}_{r+1,j} - g_{r+1,j}(\delta)\}$.
12:     **if** $|A_{r+1}| = 1$ **then**
13:         Play the only action in the remaining rounds.
14:     **end if**
15: **end for**

**Theorem 8.** *For any $\delta \in (0,1)$ and any $\theta \in \Theta$,*

$$R_T(\theta) \le (\rho(\Sigma)D)^{1/3}(\sigma T)^{2/3} \cdot 7\sqrt{6\log(2KT/\delta)} + 125\sigma^2 K^3/D + 13K^3 D$$

*with probability at least $1 - \delta$ if $\Sigma$ is weakly observable, while*

$$R_T(\theta) \le 2KD + 80\sigma\sqrt{\kappa(\Sigma)T \cdot 6\log K \log \frac{2KT}{\delta}}$$

*with probability at least $1 - \delta$ if $\Sigma$ is strongly observable.*

**Theorem 9** (Problem-dependent upper bound)**.** *For any $\delta \in (0,1)$ and any $\theta \in \Theta$ such that the optimal action is unique, with probability at least $1 - \delta$,*

$$R_T(\theta) \le \frac{1603\rho(\Sigma)D\sigma^2}{d_{\min}^2(\theta)}\left(\log(2KT/\delta)\right)^{3/2} + 14K^3 D + 125\sigma^2 K^3/D$$

$$+ 15\left(\rho(\Sigma)D\sigma^2\right)^{1/3}\left(125\sigma^2/D^2 + 10\right)K^2\left(\log(2KT/\delta)\right)^{1/2}.$$

**Remark 10.** Picking $\delta = 1/T$ gives an $O(\log^{3/2} T)$ upper bound on the expected regret.

**Remark 11.** Note that Algortihm 2 is similar to the UCB-LP algorithm of [7], which admits a better problem-dependent upper bound (although does not achieve it with optimal problem-dependent constants), but it does not achieve the minimax bound even under strong observability.

## 5 Conclusions and Open Problems

We considered a novel partial-monitoring setup with Gaussian side observations, which generalizes the recently introduced setting of graph-structured feedback, allowing finer quantification of the observed information from one action to another. We provided non-asymptotic problem-dependent lower bounds that imply existing asymptotic problem-dependent and non-asymptotic minimax lower bounds (up to some constant factors) beyond the full information case. We also provided an algorithm that achieves the asymptotic problem-dependent lower bound (up to some universal constants) and another algorithm that achieves the minimax bounds under both weak and strong observability.

However, we think this is just the beginning. For example, we currently have no algorithm that achieves both the problem dependent and the minimax lower bounds at the same time. Also, our upper bounds only correspond to the graph-structured feedback case. It is of great interest to go beyond the weak/strong observability in characterizing the hardness of the problem, and provide algorithms that can adapt to any correspondence between the mean payoffs and the variances (the hardness is that one needs to identify suboptimal actions with good information/cost trade-off).

**Acknowledgments** This work was supported by the Alberta Innovates Technology Futures through the Alberta Ingenuity Centre for Machine Learning (AICML) and NSERC. During this work, A. György was with the Department of Computing Science, University of Alberta.

## Footnotes

[1]Tilde denotes order up to logarithmic factors.

[2]To see this, notice that the error of identifying the optimal action decays exponentially with the number of rounds.

[3]Recall that $\theta_i \in [0, D]$.

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
