[Reviews · NeurIPS 2015]

Submitted by Assigned_Reviewer_1

The authors consider a novel partial information stochastic learning problem where the learner's action decides the variance of the observed Gaussian payoff. This setting covers full information, fully bandit, and partial monitoring settings. The main result is problem-dependent lower and upper bounds, which recover the previously known bounds in special cases. Theorem 1 states that any algorithm with a worst-case regret bound also has a generic problem-dependent lower bound. Algorithm 1 improves upon Magureanu et al.'s algorithm for the Gaussian payoff case, and eliminates the need for knowing d_min in advance.

Overall, the paper is well written and the math is sound. The assumptions are somewhat strong, but the paper provides a sound starting point for future work to build upon.

- Theorem 1. the notations get confusing. You might want to give name for the functions f, g, C, etc. and explain them in an English sentence. e.g., $f$ is the infimum regret over all distributions whose KL divergence is upper bounded in terms of the KL divergence between \theta,\theta'. You may want to separate the arguments over which you'll later take inf/sup over, e.g. f(\theta ; q, \theta'). - Is "generalized full information" (line 159) the same as "asymptotically full information" (line 110)? - Since we have a strong distribution assumption and the algorithm knows the feedback structure, is it feasible to obtain a regret bound in terms of the "goodness" of the sample payoffs?

Summary: The authors consider a novel partial information setting. They prove problem-dependent lower bounds, and provide algorithms that achieve the matching upper bound, which are new and interesting results.

Submitted by Assigned_Reviewer_2

The authors consider a sequential decision problem in stochastic environment with Gaussian payoffs and Gaussian side observations. In each time step a learner chooses an action from a fixed finite set of actions, receives a random reward associated with the action, and observes a feedback from other actions. All the observation are Gaussian random variables with fixed expectations and variance depending on a chosen action.

The setting includes several previously studied feedback models: full information, bandit, and partial monitoring with feedback graphs (previously studied in adversarial environment).

The setting of the paper is very interesting and naturally extends previous settings, which provides tools to capture problems with complicated feedback structure.

Pros of the paper:

The main contribution of the paper are lower bounds on the regret for any algorithm. The authors provide non-asymptotic problem-dependent lower bound, which recovers asymptotic problem-dependent lower bound, and finite-time minimax lower bound.

The authors proposed two algorithms and their finite-time analysis. The first algorithm achieves upper bound which matches asymptotic problem-dependent lower bound (up to constants) and the second algorithm achieves upper bound which matches minimax lower bound (up to logarithmic factors).

Cons of the paper:

Even though the setting provides rich feedback structures, the upper bounds for the algorithms are analysed only for a special case of uniform variance, where the variance terms are equal to either sigma or infinity (feedback graphs). In particular, the proofs of Theorem 6 and

Theorem 8, provided in appendix B and C

(proved using standard tools) are only considering this special case and it is unclear if the approach would extend to the general case which seems more interesting.

However, my main objection is that the paper lacks intuition behind the proofs (especially for the lower bound). A good example is Section 3.2.1.: There is no reason given why are the quantities defined as such. What does the set of the reasonable items mean? Where are the definitions for m_i^+ and m_i^- coming from? The whole Section 3.2.1. is extremely hard to parse and features almost no explanations or relations to the studied problem.

With regards to the lower bound, it is also unclear whether this is indeed the difficult case, since the upper-bound for the nonuniform case is not given.

Furthermore, there is a significant amount of mistakes, including wrong or missing indices. Examples: assumption (iii) in section 2, simplex using set C_N^T as upper index insection 3.1), missing terms (e.g. definition of C_T in section 2.1)). Combined with missing descriptions of the ideas, this makes it even harder to

to follow the paper.

Conclusion:

Even though the results in the paper are substantial, the paper needs significant changes to eliminate mistakes, provide intuition behind (mostly the lower bound) proofs, improve clarity and readability.

Therefore in my opinion, the paper is not ready to be published in the current form.
Summary: Even though the authors provided interesting results, especially the lower-bounds, the paper contains significant amount of mistakes and missing intuitions which contributes to the fact, that it is hard to follow the ideas in the paper. Therefore, I suggest a major rewrite.

Submitted by Assigned_Reviewer_3

The setup is quite interesting, as it is an attractive generalization of the graph-structured feedback model, but most of the bounds are hard to interpret and/or are not tight.

For example, it is not clear whether for every (or most, or a significant portion of) element q \in \Delta^{C_N^T} one can construct an algorithm with q as the distribution of the number of arm pulls. As the bounds are typically expressed in terms of this and similar other notions, it is hard to relate them to the actual goal: to derive tight bounds on the regret of optimal algorithms. Showing that the derived results lead to optimal minimax bound does not imply that they are tight in the distribution dependent sense.

On the other hand, the bounds that are expressed in a way that is easier to interpret, are clearly not tight. The lower and upper bounds differ by some log T factors, whereas the usual distribution dependent bounds in the topic are themselves typically logarithmic in T. None of the known results are recovered for special cases.

The paper would be a nice contribution, but these issues should be taken care of. ----------------------------------------------------------------------------------- The rebuttal has helped me understand the results better, and have changed my opinion on them. However, the presentation issues are still serious, and should be taken care of. For example, the paper should include a thorough discussion about the complexity parameter (with an emphasis on the results in 3.2.2 in relation to (1)) and how it comes up in the analysis (i.e., the proof ideas) in order to make the results more accessible for the reader.
Summary: The setup is quite interesting, but most of the bounds are hard to interpret and/or are not tight.

Submitted by Assigned_Reviewer_4

The paper presents a nice model in which we have a multi-armed bandit setting but where some information is shared across arms. The "some" is given by assuming a known correlation structure on information given about each arm on each round. The paper does seem to achieve some nice results, showing that a b*logT regret bound is possible, and they give a description as to what this value of b should be. Upon more careful inspection, however, I found it very difficult to understand exactly what this value b is, as it has a very involved definition and there seems to be little guidance on how to understand this quantity (it's defined through an inf). I also got confused regarding the claimed O(log^{3/2} T) upper bound for a problem-dependent algorithm. Isn't O(log^{3/2} T) worse than O(log T)? Why should a problem-dependent bound be more difficult?

My sense is that the paper was quickly written, as I found several other mistakes as well. For example, the definition of C_N^T contained a sum over.... nothing, as far as I can tell. And the definition of C_\theta contained a \sigma^2 quantity, but so far we have only been given quantities \sigma_{i,j} (with subscripts).

I would be more enthusiastic about the work if I had a sense that there were major breakthroughs here. At the moment I'm just not sure, but could be convinced by another reviewer or through feedback.
Summary: I like the setting, and I like the connection to existing bandit techniques. I'm a bit fuzzy on what the paper achieves, and the writeup seems rushed.

Submitted by Assigned_Reviewer_5

The paper introduces a variation of the stochastic bandit framework.

For each handle i = 1,...,K there is unknown mean payoff \theta_i and known st.d. \sigma_{ii}. Upon choosing handle i, the learner receives a gain with the Gaussian distribution N(\theta_i,\sigma_{ii}).

The novelty in the approach is that the information on the outcomes of alternative actions comes as random variables of the following kind. The possible gain of choosing handle j is as a Gaussian random variable with the distribution N(\theta_j, \sigma_{ij}), where \sigma_{ij} are known.

If \sigma_{ij} is small, then choosing handle i provides a lot of information on handle j. If \sigma_{ij} is large, little information is obtained.

The paper carries out a thorough mathematical investigation of this framework and develops algorithms with tight bounds.

The paper provides an important contribution to the theory of stochastic bandits, which have become popular again recently.

Minor comments:

1. Page 1, lines -1 -- -3: This sentence is very hard to parse. I think "whose" and "its" refer to the same object and the same word should be used.

2. Page 2, line -12:

\theta\in [0,D] -> \theta\in [0,D]^K

3. Page 2, line -12: \sigma_{ij}\le \sigma ->

\sigma_{ii}\le \sigma (provided I understood the discussion correctly).
Summary: The paper introduces a variation of the stochastic bandit framework and carries out a thorough investigation.

Author Feedback
Author rebuttal: At a high level, the reviewers are concerned by the quality of presentation (typos, lack of intuitive explanations), some of them are concerned about the importance of the results and the tightness of both the upper and lower bounds, and some ask for more general results.

- Improving presentation quality seems easy to address in the final submission: We will improve the presentation by correcting all the typos remaining and adding explanations as requested, while staying within the page limit (we will extract the intuitive explanations from the proofs, e.g., m_i^+ and m_i^- come from the minimum sample size required to distinguish between an arm and its perturbed version for some worst-case perturbation).

- Importance and tightness of the results:
* *Finite time problem dependent* lower bounds are completely novel in the literature for any stochastic online learning problem (including the standard bandit setting) and we believe the techniques we used can be useful to derive such lower bounds for other settings, too. We know of nothing similar before.
* Our bounds are essentially tight in several known cases:
+ Lower bound (Sections 3.2.2 and 3.2.3): our lower bound can reproduce both asymptotic problem dependent and minimax lower bounds, so we argue that it is tight. (We have a regrettable typo in Sec 3.2.2: "Theorem 1" everywhere here should be "(1)".)
+ Upper bounds: For the uniform variance model, we achieve state of the art results by the two algorithms: Algorithm 1 gets the O(log T) problem dependent bounds and improves on Magureanu et al as noted by Rev#1, while Algorithm 2 achieves the minimax bounds (T^{1/2} or higher) up to (usually thought to be negligible) log factors. To our knowledge, Alg. 2 is the first algorithm to achieve in the strongly observable setting simultaneously a minimax bound and a problem dependent polylog bound.

- Concerning generality, the uniform variance model can be trivially extended to the uniform subgaussian case, which can be regarded as the stochastic version of [5], interesting on its own. Further, the case considered already has a quite rich structure, presenting significant algorithm analysis and design challenges.

Next we address the reviewers' remaining individual concerns:

Reviewer_1:

- Confusing notation in Theorem 1: While Section 3.1 already has some explanation, we will add more space permitting.

- Is "generalized full information" (line 159) the same as "asymptotically full information" (line 110)?
Yes, we'll keep only the first.

- [..] is it feasible to obtain a regret bound in terms of the "goodness" of the sample payoffs?
We suppose you are talking about data-dependent bounds. We have not considered this.

Reviewer_2:

General case: We are not sure whether our algorithms can be generalized.

Reviewer_3:

- Page 2, line -12: \sigma_{ij}\le \sigma -> \sigma_{ii}\le \sigma.
No. We are considering the more general partial monitoring case, where an action may not be self observable (sigma_{ii} -> infty). This assumption says the reward of each action j can be effectively observed by at least one action (may or may not be j itself).

Reviewer_4:

Dealing with unknown variances is an interesting future work. However, the known variances model is general enough to cover several interesting existing models and raises interesting new challenges that go beyond the current state of the art.

Reviewer_5:

- O(log^{3/2} T) worse than O(log T)?

For minimax bounds, extra logs are negligible. See discussion at the top.

- \sigma is introduced in Line 108, before the definition of C_\theta.

Reviewer_6:

- It is not clear whether for every (or most, or a significant portion of) element q \in \Delta^{C_T^N} one can construct an algorithm with q as the distribution of the number of arm pulls.

The algorithm that pulls action i C_i times where C\in C_T^N is randomly chosen at time zero s.t. Prob{C=c}=q(c) [with some abuse of notation] does this.

- Showing that the derived results lead to optimal minimax bound does not imply that they are tight in the distribution dependent sense.

See the discussion at the top.

- On the other hand, the bounds that are expressed in a way that is easier to interpret, are clearly not tight.

Actually the tightness of our lower bound is shown based on the relaxed version (Corollary 3, Section 3.2.1-3.2.3). The asymptotic bound is shown to be tight by our Algorithm 1 and the minimax bound matches existing results in [5]. So we think even the relaxed version in Corollary 3 is still tight in our compared cases.

- The lower and upper bounds differ by some log T factors, whereas the usual problem dependent bounds in the topic are themselves typically logarithmic in T. None of the known results are recovered for special cases.

See the discussion above on the importance of the results (essentially, Alg 1 gets the log T bounds, see Corollary 7).